# Olive Leaves as a Source of Anticancer Compounds: In Vitro Evidence and Mechanisms

**DOI:** 10.3390/molecules29174249

**Published:** 2024-09-07

**Authors:** Heloisa Rodrigues Pessoa, Lilia Zago, Graziana Difonzo, Antonella Pasqualone, Francesco Caponio, Danielly C. Ferraz da Costa

**Affiliations:** 1Laboratory of Physiopathology and Biochemistry of Nutrition, Nutrition Institute, Rio de Janeiro State University, Rio de Janeiro 20550-013, Brazil; mrs.pessoa@gmail.com (H.R.P.); lilia.zago@gmail.com (L.Z.); 2Department of Soil, Plant and Food Sciences, Food Science and Technology Unit, University of Bari Aldo Moro, Via Amendola 165/A, I-70126 Bari, Italy; graziana.difonzo@uniba.it (G.D.); antonella.pasqualone@uniba.it (A.P.); francesco.caponio@uniba.it (F.C.)

**Keywords:** olive leaves, phytochemicals, anticancer activity

## Abstract

Olive trees not only produce olives but also generate a substantial amount of waste and by-products, including leaves, pomace (the solid remains after pressing olives for oil), and wastewater from the olive oil-making process. The waste products, particularly the leaves, contain bioactive compounds, especially phenolic compounds, known for their health benefits, such as high antioxidant potential and the ability to reduce inflammation. These compounds have shown promise in preventing and treating cancer. This review, based on in vitro evidence, provides a detailed description and discussion of the mechanisms through which these compounds from olive leaves can prevent development, the ways they might act against cancer cells, and their potential to increase the sensitivity of tumor cells to conventional anticancer therapy. The possible synergistic effects of these compounds suggest that olive leaf extracts may offer a promising approach for cancer treatment, compared with isolated compounds, thus providing novel possibilities for cancer therapy.

## 1. Introduction

The olive tree (*Olea europaea* L.), belonging to the *Oleaceae* family, is a subtropical evergreen tree [1]. Its distribution in the Mediterranean Basin has significant social, economic, and ecological implications. This tree has adapted to grow under different climatic conditions, altitudes, soils, and temperature regimes [2]. The cultivation of olive trees and the production of olive oil result in large amounts of solid waste and liquid effluents, including olive leaves, pomace, and olive oil mill wastewaters. The use of solid residue holds significant economic and social importance for the Mediterranean area, as it accumulates in large volumes, raising environmental concerns [3]. Due to the increasing global demand for olive oil over the past 20–30 years, interest in olive oil production has expanded to regions and countries outside the Mediterranean Basin, such as Australia, China, India, and South America [4].

Olive leaves represent an agricultural waste obtained during the harvesting of olive trees. Pruning alone produces approximately 25 kg of waste per olive tree annually, consisting of branches and leaves. Moreover, a large number of olive leaves are also discarded during the olive drupes washing process at the beginning of the olive oil production chain [3].

Olive leaves are abundant in various known phenolic compounds, broadly categorized into (i) secoiridoids (including oleuropein and oleuropein-aglycone), (ii) simple phenols (such as hydroxytyrosol—HT and tyrosol), and (iii) flavonoids (such as rutin and luteolin-7-glucoside) (Figure 1) [5]. Consequently, this agro-industrial waste is a potential inexpensive, renewable, and abundant source of phenolic compounds characterized by numerous health benefits [6]. The olive leaves have been used to obtain extracts, particularly isolated compounds, by the pharmaceutical and cosmetics industries. Notably, in some regions, they are consumed as olive tea, prepared from either fresh or dried leaves. Olive leaves contain several bioactive compounds with antimicrobial, antioxidant, hypoglycemic, anti-hypertensive, hypocholesterolemic, and anti-inflammatory properties [7]. Additionally, they exhibit anticancer potential by preventing tumor suppression in different cancer models [8].

Statistics highlight the importance of designing measures to mitigate cancer incidence or impede the progression of neoplasms. According to GLOBOCAN 2020, it is projected that there will be 28 million new cancer cases annually worldwide by 2040. The four most common types of cancer globally are female breast, lung, bowel (including anus), and prostate cancers, accounting for more than 43% of all new cases [9].

In the context of cancer, phytochemicals, such as those found in olive leaves, have been studied for their antioxidant, pro-apoptotic, anti-inflammatory, anti-angiogenic, anti-carcinogenic, and anti-metastatic properties, as well as their selective cytotoxicity to cancer cells. Consequently, these compounds have the potential to contribute to an efficient and less aggressive therapy [10]. Additionally, olive leaf extracts may be more beneficial than isolated constituents, as a bioactive individual component can alter its properties in the presence of other compounds within the extracts. For this reason, olive leaf extracts are of particular interest for their therapeutic effects [6].

The purpose of this paper is to review the antitumor activities of olive leaves and their bioactive compounds. We first briefly stated the antitumor activity of the bioactive compounds of olive leaves, particularly major phenolic compounds, and focused on current research investigating their effects on cancer development and the possible mechanisms behind them, including the combination of phenolic compounds with anti-cancer drugs. Furthermore, special emphasis was placed on new evidence regarding the effects of olive leaf extract on tumor cells, aiming to provide a perspective on the antitumor potential of olive leaf extract and future research directions for the extract.

## 2. Olive Leaves Composition and Phytochemicals

Bioactive compounds, or phytochemicals, are substances naturally present in plant-origin foods and plants that, while not classified as essential nutrients, can exert bioactive effects on human health [11,12].

Both the products and by-products of olive trees contain bioactive substances. Olive leaves are a rich source of bioactive compounds, mainly phenolic compounds. Phenolic compounds constitute an important class of secondary metabolites produced by plants in response to environmental stimuli [13].

Olive leaves are oblong, measuring 5–10 cm in length and 1–3 cm in width, with a silvery green appearance. When consumed, these have a sharp, bitter taste. The main phenolic compounds found in olive leaves are oleuropein, hydroxytyrosol, elenolic acid, and tyrosol. Regarding flavonoids, luteolin-glucoside isomers have been detected [14,15]. The composition of these compounds is influenced by factors such as cultivar, region, harvesting period, and geographical location [16,17]. Olive leaves also contain a considerable amount of pentacyclic triterpenoids, with oleanolic acid being the predominant component, ranging from 3.0 to 3.5% up to 3.98% on a dry basis. The concentration of alpha-tocopherols in olive leaves can vary around 10.12 µg/g (dry basis) to 82.37 µg/g (dry basis) and can reach up to 284.6 μg/g (dry basis), whereas young leaves tend to have higher contents of oleuropein, ligstroside, and flavonoid aglycones [17,18].

The phenolic groups predominantly found in olive leaves are (i) secoiridoids, such as oleuropein and oleuropein-aglycone, which are characteristic of the *Oleaceae* family, (ii) flavonoids, such as rutin and luteolin-7-glucoside, and (iii) simple phenols, such as hydroxytyrosol and tyrosol. Among these, oleuropein and hydroxytyrosol are abundant in olive leaves [18]. However, mature olive-leaf extracts may contain higher levels of verbascoside isomers and glycosidic forms of luteolin, whereas young leaves tend to have higher contents of oleuropein, ligstroside, and flavonoid aglycones [3].

Oleuropein is the most abundant bioactive compound in olive tree products. This secoiridoid is the primary bioactive component present in the olive tree, which confers the main bitter taste and provides resistance to the development of oil rancidity [19]. The concentration of oleuropein in olive leaves has been reported to be in the range of 6 to 9% on a dry basis (60–90 mg/g dry matter in leaves) [17]. However, this may vary depending on the cultivar and planting conditions [20]. Oleuropein, a secoiridoid glycoside belonging to the class of coumarin components present in olive leaves, is characterized by an ester linkage of elenolic acid glucoside and 2-(3′,4′-dihydroxyphenyl) ethanol (hydroxytyrosol) [21,22]. In its chemical structure, oleuropein contains an ortho-diphenolic group capable of scavenging ROS through hydrogen donation and stabilizing oxygen radicals with an intramolecular hydrogen bond. In particular, an *o*-diOH substitution confers a high antioxidant property, whereas single hydroxyl substitutions, e.g., tyrosol, provide none [23] and the ability of oleuropein to chelate metal ions such as iron enhances its antioxidant activity [24].

This antioxidant potential is generally attributed to the main health properties related to oleuropein. As reported in a recent review, oleuropein has antioxidative, antimicrobial, antiviral, cardioprotective, antihypertensive, and anti-inflammatory hallmarks, as well as hypocholesterolemic and hypoglycemic activities, together with the lipid metabolism enhancement effect, in addition to the ability to exert as a natural anticancer and pro-oxidant agent [3,23,25]. Despite the effects attributed to the component in isolation, using olive leaf extracts without isolating the major constituent may be recommended to achieve the best health properties. This is due to the synergy of all bioactive compounds present in the extracts, which most likely affects their absorption and bioavailability [26].

Hydroxytyrosol (3,4-dihydroxyphenylethanol), a phenolic alcohol, has been reported to be present at around 2.28 mg by 100 g leaf extract. This compound is a potent antioxidant derived from the hydrolysis of oleuropein [17,18]. Hydroxytyrosol has a simple molecular structure, making it easy for the human body to assimilate, with high bioavailability. It reaches blood plasma in 15 or 20 min and is eliminated 6–8 h later by the renal or digestive system, thus not presenting accumulation or toxicity issues. It is an amphipathic, water-soluble, and fat-soluble molecule because it contains a lipophilic end and a hydrophilic end, which makes it a good transporter of substances throughout the human body, allowing it to penetrate the cellular membrane more easily. Because of its structural and molecular properties, hydroxytyrosol intake offers a wide range of benefits for the organism [27,28].

Hydroxytyrosol demonstrates the potential to defend against chronic diseases due to its high antioxidant potential. This efficiency is primarily attributed to the presence of the o-dihydroxyphenyl moiety. Its main role is acting as a chain breaker, donating a hydrogen atom to peroxyl radicals (ROO*), serving as a free radical scavenger, and acting as a metal chelator. Moreover, it enhances antioxidant protection by reinforcing the natural defense mechanisms against oxidative stress and activating distinct cellular signaling pathways [29].

Tyrosol (2-(4-hydroxyphenyl)-ethanol, Tyr) is typically found in olive leaves in trace amounts, approximately 0.0007 mg/g leaves on a dry basis [5,17]. Tyrosol maintains its antioxidant activity even under challenging conditions and is a stable compound, making it less susceptible to autooxidation compared with other polyphenols. However, it undergoes extensive metabolism in the human body, leading to the poor bioavailability of its metabolites. Tyrosol is rapidly absorbed and excreted via the kidneys within 8 h after oral administration [17].

## 3. Anticancer Properties of the Main Olive Leaves Compounds

The main phenolic compounds found in olive leaves, oleuropein and hydroxytyrosol, have been reported in the literature to exhibit anticancer properties both in isolation and as pharmaceutical adjuvants. These compounds have been investigated for their in vitro anti-tumor effects using various cellular models, including breast cancer, melanoma, cervix, ovarian, colon, colorectal adenocarcinoma, hepatocellular carcinoma, osteosarcoma, squamous cell carcinoma of the head and neck, neuroblastoma, liquid cancers, leukemias, myelomas, and lymphomas (Table 1).

### 3.1. Anticancer Effects of Oleuropein In Vitro

Numerous studies have demonstrated the capability of oleuropein to induce apoptosis in cancer cells. Specifically in breast cancer cells, researchers have observed a decrease in cell viability upon exposure to oleuropein, with this effect being both time- and concentration-dependent. In a study by Han et al. (2009), an ethanolic extract of olive leaf was preliminarily screened, showing an antiproliferative effect on breast cancer cells. The authors hypothesized that the key compounds within this extract were responsible for this effect. Through their investigation, they used isolated hydroxytyrosol and oleuropein and showed that oleuropein (at 200 μg/mL) reduced the cell viability of MCF-7 breast cancer cells. Moreover, significant morphological changes indicative of apoptosis was observed, including cell shrinkage, chromatin condensation, and the formation of apoptotic bodies. The study further revealed that oleuropein (12 h, with 200 μg/mL) promotes apoptosis by activating caspases and arresting the cell cycle at the G1 phase (12, 24, and 48 h with 100 μg/mL) [30].

In a subsequent study by Bulotta et al. (2011), the effects of different concentrations of oleuropein and its acetylated derivatives on MCF-7 cell viability were investigated. Only the oleuropein derivatives exhibited growth inhibition at 100 μM, with the peracetylated aglycone showing the most pronounced effect. The cytotoxic impact of the peracetylated compounds was notably higher than that of the control, leading to a significant reduction in cell viability. These findings suggested that the peracetylated compounds (peracetylated aglycone and peracetylated hydroxytyrosol) induced an antiproliferative effect by halting cell cycle progression in the G2/M phase in MCF-7 cells [21].

Hassan et al. (2013) explored the gene expression levels of anti-apoptotic *Bcl-2*, pro-apoptotic *Bax*, and *p53* in MCF-7 breast cancer cells following treatment with 200 µM oleuropein. The results showed a significant upregulation of the Bax gene by 0.6 folds compared with untreated cells and those treated with 100 µM oleuropein. Additionally, oleuropein was found to induce apoptosis in MCF-7 cells through a *p53*-dependent pathway, with treatment leading to a substantial increase in *p53* gene expression (by 2.5 and 3.5 folds compared with the untreated cells) [31].

In a more recent study by Choupani et al. (2019), the effects of oleuropein and oleuropein-doxorubicin (DOX) treatment on MCF-7 cells were compared. Oleuropein alone demonstrated a dose-dependent inhibition of cell migration and a cytotoxic effect, with an IC_50_ value of 588 µg/mL. In contrast, the IC_50_ value for DOX treatment was 0.48 µM. The combined treatment of DOX (0.05 to 1 µM) with oleuropein (600 µg/mL) resulted in a significant enhancement of the cytotoxic effect of DOX, lowering the calculated IC_50_ for DOX by 6.40 times. Furthermore, the combination treatment increased the ratio of late/early apoptosis in comparison with DOX treatment alone [32].

The authors also evaluated the expression levels of MMP-2 and MMP-9, members of a family of zinc-dependent proteolytic enzymes capable of remodeling the extracellular matrix (ECM) and degrading proteins. Their unregulated expression is related to angiogenesis, migration, invasion, and metastasis. These enzymes were significantly downregulated in oleuropein-treated breast cells. An increase in E-CAD expression of 13.9-fold was also reported; E-CAD function is generally disrupted in tumor invasion and metastasis. This confirms the antimigratory effect of oleuropein. On the other hand, oleuropein downregulated ZEB1, which decreased the formation of the SIRT1/ZEB1 complex, an E-CAD suppressor complex, subsequently leading to an increased expression of E-CAD. When combined with the downregulation of MMP-2 and MMP-9 by oleuropein, this effect further suppresses the epithelial-mesenchymal transition (EMT), a primary process that enables tumor cells to migrate from their microenvironment to distant locations. Furthermore, oleuropein, by upregulating p53 as an upstream SIRT1 expression regulator, can suppress SIRT1 expression, which helps maintain the epithelial phenotype and attenuates invasion and metastasis [32].

According to Messeha et al. (2020), oleuropein decreases the cell viability of triple-negative breast cancer cells (MDA-MB-231 and MDA-MB-468 cells). The antiproliferative effects of the compound in both cell lines were demonstrated in a dose- and time-dependent manner. A highly significant cytotoxic effect (*p* < 0.0001) was identified in MDA-MB-231 (100 to 700 µM) and in MDA-MB-468 (100 to 400 µM). The data suggested apoptosis as the anticipated primary mode of cell death in oleuropein-treated MDA-MB-468 cells. However, MDA-MB-231 exhibited higher resistance to apoptosis and tended to undergo necrosis. Oleuropein also modified the expression of many apoptosis-involved genes. In MDA-MB-468 cells, there was a dramatic increase in two members of the caspase family (CASP1 and CASP14), proapoptotic genes (*GADD45A*, *BNIP2*, *BNIP3*, *BID*, and *BCL10*), and two members of the TNF receptor superfamily (FADD and TNFRSF21), in addition to *CYCS* and *CFLAR* genes. These genes augment apoptosis by triggering intrinsic, extrinsic, or both apoptotic pathways. For MDA-MB-231 cells, apoptosis was mainly induced by the downregulation of the antiapoptotic gene *TNFRSF11B* and survivin (BIRC5) and the minor upregulation of CASP4, notwithstanding alterations in genes resisting apoptosis induction [34].

Similar results were observed by Liu et al. (2019), where MDA-MB-231 and MCF-7 cells were exposed to oleuropein (0 to 100 μM) for 24, 48, and 72 h. Oleuropein significantly decreased the viability of MDA-MB-231 cells (ER-negative breast cancer) in a dose- and time-dependent manner, while the MCF-7 cells were more resistant to the treatment. The wound healing assay showed that the migration of MDA-MB-231 cells was significantly inhibited upon oleuropein treatment. After 72 h of treatment, the accumulation of MDA-MB-231 cells in the sub-G1 phase dramatically increased with rising concentrations of oleuropein. Furthermore, the percentage of apoptosis in MDA-MB-231 cells significantly increased in a dose-dependent manner. This was accompanied by a significant increase in the cleavage of PARP and caspase-3/7 activities, suggesting that oleuropein can induce substantial apoptosis in MDA-MB-231 cells. The study also showed that oleuropein significantly inhibited NF-κB activation at 36 and 48 h, suppressing the NF-κB signaling cascade [33].

Samara et al. (2017) developed semi-synthetic analogs of oleuropein by modifying the structure of the natural compound through chemical reactions, resulting in 51 new analogs divided into two sub-groups of aryl and alkyl ester derivatives. Initially, the authors investigated the cytotoxicity of oleuropein from olive leaf extracts enriched with oleuropein in several cellular models of cancer [22].

In MCF-7 cells [breast, estrogen receptor alpha positive (ERα+)] and SKBR3 cells [breast, ERα negative (ERα−)], a cytotoxic effect was observed. The calculated IC_50_ ± SD (in μg/mL) for MCF-7 cells was 91.67 ± 14.43 (169.70 μM) and 120.00 ± 5.00 for oleuropein and enriched olive leaf extract, respectively. For SKBR3 cells, the respective values were 86.67 ± 15.28 (160.44 μM) and 104.30 ± 6.03. Similar assays were performed on FM3 (melanoma), HCT-116 (colon), and HeLa (cervix) cells. The IC_50_ ± SD (in μg/mL) for treatment with oleuropein were 148.30 ± 2.89 (274.53 μM), 100.00 ± 13.23 (185.12 μM), and 143.30 ± 15.28 (265.28 μM), respectively, while with enriched olive leaf extract, they were 240.00 ± 10.00, 174.30 ± 8.15, and 165.00 ± 5.00. This suggests that oleuropein likely contributes to the overall cytotoxic activity of the extract. However, oleuropein and the enriched olive leaf extract exhibited overall weak cytotoxicity compared with the chemotherapeutic doxorubicin [22].

To increase the antitumor potential of the molecule, Samara et al. (2017) developed synthetic analogues of oleuropein. The most promising analog compounds lacked the hydroxytyrosol moiety. One of the synthetic forms of oleuropein (Analogue 24) efficiently and selectively killed cancer cells without causing severe toxicity in peripheral blood lymphocytes and displayed potent in vivo activity against melanoma, retarding tumor growth and stimulating antitumor immune responses primarily by inducing the in vivo expansion of melanoma-reactive effector T cells. The analogue 24 has a cetyl ester (sixteen carbons) at position 7 and a methyl ester at position 11, which the original compound (oleuropein) does not have [22].

Scicchitano et al. (2023) described that at lower doses (100 to 200 μM) oleuropein acts as an antioxidant and iron chelator in oxidative stress conditions induced by erastin in ovarian cancer cells, counteracting the cytotoxic effects induced by erastin and reverting mitochondrial dysfunction. The study also found that low doses of oleuropein reduce erastin-mediated cell death and decrease the levels of intracellular ROS and LIP in ovarian cancer cells treated with erastin. Additionally, the authors used the CM-H2DCFDA flow cytometry assay to measure oxidative stress and evaluated mitochondrial dysfunction based on mitochondrial ROS and GPX4 protein levels. An increased amount of the ROS scavenging enzyme GPX4, along with a consistent reduction in mitochondrial ROS, was observed, confirming a reduction in oxidative stress in this model [19].

High doses of oleuropein also show antiproliferative and pro-apoptotic activity in ovarian cancer cells. This study reports that high doses of oleuropein mediate HEY ovarian cancer cell growth inhibition by promoting cell cycle arrest and apoptosis. Specifically, treatment with 400 μM oleuropein for 24 h leads to a significant decrease in S-phase (from 36.63% to 12.97%; *p*-value = 0.03) in parallel with an increase in both the subG1 population (from 0% to 3.57%; *p*-value = 0.01) and the G2/M population (from 18.3% to 46.1%; *p*-value = 0.005) in HEY cells. Therefore, oleuropein promotes cell cycle arrest in the subG1 and G2/M phases of the cell cycle in HEY cells. Treatment with 400 μM oleuropein for 24 h increased the percentage of apoptotic cells from 2.3% to 54.8% in HEY-treated cells compared with control cells. On the other hand, low doses of oleuropein impair cell viability without affecting the mortality of cells and also decrease the LIP and ROS levels [19].

Oleuropein has no effect on hepatocellular carcinoma cell viability as a sole treatment agent. Oleuropein at concentrations in the range of 20 to 100 μM led to morphological alterations, reduced cell growth, and cell death in HepG2 cell lines. These cells showed an increased expression of caspase and pro-apoptotic proteins of the Bcl-2 family and suppression of the PI3K/AKT signaling pathway, an important mediator of apoptosis induction [35].

According to Katsoulieris (2016), the treatment of HepG2 cells with various concentrations of isolated oleuropein for 24 h did not affect cell viability, as evident from unchanged MTT reduction. However, oleuropein partially restores HepG2 cell viability in the presence of induced oxidative stress. Concomitant treatment of PQ (a toxic herbicide that induces oxidative stress and cell death) with oleuropein partially restores HepG2 cell viability and cytoskeletal integrity. The protective effects of oleuropein on cellular viability involve a significant reduction in necrotic cell death levels, accompanied by the suppression of Casp-3 cleavage. PQ-induced necrosis appears to follow an apoptosis event in the combined treatment group (PQ + OP), indicating a shift from the necrotic pathway to apoptosis. Oleuropein enables HepG2 cells to better cope with PQ insult and survive longer, prior to entering a more regulated cell death phase, rather than experiencing instant necrosis as seen in oxidative stress-treated cells [36].

Concomitant treatment with phytochemicals and chemotherapeutic drugs has been the subject of research. There is evidence that oleuropein (200 μM) potentiates the anti-tumor activity of cisplatin (50 μM) against HepG2 cells. The gene expressions of nerve growth factor (NGF) and matrix metalloproteinase-7 (MMP-7) were evaluated. The activity of MMP-7 is essential for the conversion of pro-NGF, which promotes apoptosis, to NGF, which stimulates survival and differentiation. The dysregulation of growth factor signaling plays a significant role in cancer development and progression, including the metastasis and angiogenesis of tumors. Concurrent treatment with both oleuropein and cisplatin could lead to a more effective chemotherapeutic combination against HCC compared with oleuropein or cisplatin alone, attributed to the influence of oleuropein on the pro-NGF/NGF balance via affecting MMP-7 activity without altering the gene expression of NGF [37].

Gioti et al. (2021) investigated the effect of oleuropein alone (20 μg/mL) and in co-treatment with Adriamycin (ADR) at 50 nM, an anthracycline widely used as a chemotherapeutic agent, in MG-63 human osteosarcoma cells. Oleuropein exhibited cytotoxic effects in MG-63 cells (IC_50_ = 22 μg/mL ± 3.6). It was also observed that the treatment of MG-63 cells with 20 μg/mL of oleuropein for 24, 48, and 72 h produced a limited increase in the cell distribution in the G0/G1 (Gap 0/Gap 1) phase. Additionally, the co-treatment with ADR and oleuropein enhanced the cell cytotoxicity with significantly lower ADR doses than the ADR treatment alone, demonstrating that oleuropein potentiates ADR’s cytotoxicity [38].

To elucidate the molecular mechanism underlying this ADR and oleuropein’s synergistic effect, Gioti et al. (2021) investigated the expression of autophagy-related genes. The gene expression profile of MG-63 cells revealed that oleuropein treatment alone strongly enhanced the expression of *ULK1*, *AMBRA1*, and *BniP3L* mRNAs, whereas LC-3 expression was greatly suppressed. This suggests that oleuropein mediates the induction of autophagy and probably mitophagy (selective degradation of mitochondria by autophagy). In the case of ADR and oleuropein co-treatment, oleuropein enhances the already established effect of ADR autophagy induction, resulting in the dismantling of the autophagic machinery. Therefore, oleuropein, as a natural bioactive compound, could serve as a potential candidate for the design of more effective adjuvant treatments for osteosarcoma patients [38].

Isolated oleuropein was also investigated regarding its ability to affect cellular behavior, specifically in terms of viability, invasion, and apoptosis, within the SH-SY5Y neuroblastoma cell line. Seçme et al. (2016) revealed a significant decrease in viability with increasing oleuropein concentrations, with an IC_50_ value of 350 μM at 48 h, in a dose-dependent manner. To understand the molecular mechanisms behind this inhibition, the expression of 84 cell cycle control and 84 apoptosis-related genes was evaluated using RT-PCR. Oleuropein led to cell cycle arrest by downregulating the expression of genes involved in cell cycle progression (*CyclinD1*, *CyclinD2*, *CyclinD3*, *CDK4*, *CDK6*) and upregulating genes that promote cell cycle arrest (*p53*, *CDKN2A*, *CDKN2B*, *CDKN1A*). This dual action hinders the cell’s ability to divide and proliferate [39].

Furthermore, oleuropein induced apoptosis by inhibiting the expression of the anti-apoptotic gene *Bcl-2* and activating pro-apoptotic genes (*Bax*, *caspase-9*, *caspase-3*). This action was confirmed by the significant increase in the apoptotic cell ratio in the presence of oleuropein, reaching 36.4 ± 3.27%. The study also investigated the effects of oleuropein on cell invasion, colony formation, and migration, and associated aspects of cancer progression. Oleuropein demonstrated a reduction in invasion and colony formation by 53.6 ± 4.71% and inhibited cell migration, as demonstrated by the wound-healing assay. According to the authors, oleuropein shows promise in combating this aggressive pediatric tumor [39].

In summary, oleuropein demonstrates a range of anticancer effects, including apoptosis induction, proliferation inhibition, and selective toxicity towards cancer cells (Figure 2). These findings suggest the potential of oleuropein as a therapeutic agent in cancer treatment.

### 3.2. Anticancer Effects of Hydroxytyrosol In Vitro

In the late 1990s, hydroxytyrosol began to be evaluated for its in vitro antitumor potential. Manna et al. (1997) employed Caco-2 human cell lines to simulate the effects of oxidative stress on the intestinal epithelium. The authors initially revealed that treatment with hydroxytyrosol and tyrosol (250 μmol/L to 500 μmol/L) showed remarkable efficacy in preventing cellular damage induced by both H_2_O_2_ and xanthine oxidase. Even at relatively low concentrations, hydroxytyrosol completely shielded the cells from oxidative stress. In contrast, tyrosol did not demonstrate any protective effect, even at higher concentrations [40]. Subsequently, Della Ragione et al. (2000) investigated the treatment of HT (50 to 100 μM) in two colorectal cell lines (HT-29 and CaCo_2_) and found that the compound was not able to induce cell death [45].

In recent decades, there have been advances in the investigation of the role of hydroxytyrosol in human colon adenocarcinoma cells. A study published in 2009 demonstrated that hydroxytyrosol (5.0 to 162.5 μM) induced a cell cycle block in the G2/M phase, effectively halting the division and proliferation of human colon adenocarcinoma cells. This action is linked to the inhibition of extracellular signal-regulated kinase (ERK) 1/2 phosphorylation, a key signaling pathway in cell growth, and a subsequent reduction in cyclin D1 expression. Cyclin D1 is a protein that regulates the cell cycle, and its decreased expression contributes to the cell cycle arrest observed. This means that hydroxytyrosol induced a block in cell growth and division, which is crucial for the development of cancer [41].

Hormozi et al. (2020) conducted their study using LS180 cells, a human colorectal cancer cell line, and treated them with varying concentrations of hydroxytyrosol (50, 100, and 150 μM) for 24 h. Hydroxytyrosol significantly increased the expression of the *CASP3* gene, which plays a crucial role in cell death, and altered the BAX:BCL2 ratio in favor of cell death. This shift is significant because BAX promotes cell death, while BCL2 inhibits it. The treatment also reduced the expression of the *NFE2L2* gene, which is associated with cancer cell survival. Furthermore, hydroxytyrosol treatment led to a notable increase in the activity of antioxidant enzymes, including catalase, superoxide dismutase, and glutathione peroxidase. These enzymes are essential for neutralizing harmful molecules that can lead to cell damage. The treatment also reduced the levels of thiobarbituric acid-reactive substances, indicating a decrease in oxidative stress, a known factor in cancer development. These findings suggest that hydroxytyrosol may enhance the expression of genes that promote apoptosis and prevent proliferation in colorectal cancer cells by bolstering the cell’s antioxidant defenses [42].

Despite these findings, a study published in 2021 investigated the antitumor action of hydroxytyrosol (5, 10, and 20 μM) on colorectal cancer cells (SW620 cell line), associated with the promotion of oxidative stress [43]. The study demonstrated that HT effectively inhibits TrxR1 activity, a protein that acts to regulate cell damage. This inhibition leads to the accumulation of ROS within the cells, a process known to induce cell death. Additionally, the study highlighted that the interaction between HT and TrxR1 is particularly dependent on the selenocysteine residue, a key component of TrxR1’s structure and function. Furthermore, HT induced apoptosis and G1/S cell cycle arrest in SW620 cells. These cellular responses are indicative of the compound’s potential to halt the uncontrolled growth of cancer cells, a hallmark of cancer. The authors suggest that using HT to target TrxR1 could be a potential strategy for treating colorectal cancer, as it not only kills cancer cells directly but also disrupts their defense mechanisms. Additionally, the compound is effective and less aggressive than traditional drugs such as auranofin.

A study published in 2023 conducted experiments using Caco-2 cells to investigate the effects of HT on DNA methylation. DNA methylation is an epigenetic mechanism crucial for regulating gene expression, and alterations in this process are commonly associated with cancer development. In this study, the cells were treated with HT (0, 5, 10, 50, 100, or 150 µM), resulting in a significant increase in global DNA methylation, indicating that HT can influence this epigenetic process. The DNA methylation analysis revealed 32,141 differentially methylated sites, known as CpGs (specific parts of the DNA that can be methylated), after HT treatment. Among these, the endothelin receptor type A (*EDNRA*) gene stood out as a potential target of HT. The *EDNRA* gene is involved in cellular processes related to cancer development and progression. The study’s identification of the endothelin receptor type A (*EDNRA*) gene as a potential target of HT highlights the specificity and complexity of its effects on cellular processes and also paves the way for the development of innovative cancer prevention strategies centered around HT and its epigenetic modulatory properties [44].

The antitumor effect of hydroxytyrosol has also been explored in hepatocellular carcinoma cells. Initially, the HepG2 cell line was treated with hydroxytyrosol (10 to 40 μM), and no changes in cell integrity or antioxidant status were verified [48]. In contrast, Martin (2010) demonstrated that HepG2 cells treated with HT (0.5, 1.0, 5.0, and 10.0 μM) led to a significant increase in the expression and activity of glutathione-related enzymes (glutathione peroxidase, glutathione reductase, and glutathione S-transferase) that play a vital role in neutralizing ROS. Furthermore, HT induced the nuclear translocation of the Nrf2 transcription factor. Nrf2 is a key player in the cellular defense against oxidative stress, as it regulates the expression of various antioxidant and detoxifying enzymes. This translocation was facilitated by the activation of two signaling proteins, protein kinase B (AKT) and extracellular regulated kinases (ERK), which are part of the PI3K/AKT and ERK pathways, respectively [49].

In Hep3B and HepG2 cell lines, hydroxytyrosol (30 to 200 μM) caused an antiproliferative effect, observed by the inhibition of the lipogenic enzyme fatty acid synthase (FAS), the induction of the cellular antioxidant system, and the reduction of cellular levels of IL-6 at treatment concentrations up to 80 μM. However, higher concentrations increased IL-6 levels [50]. It has also been demonstrated that treatment with hydroxytyrosol (1 μM and 5 μM) attenuated endoplasmic reticulum stress induced by tunicamycin in HepG2 cells [51].

Although it has been shown that different cellular models of hepatocellular carcinoma (HepG2, Hep3B, SK-HEP-1, and Huh-7) treated with hydroxytyrosol (100 to 400 μM) suppress cell proliferation, the same effect is not observed in the non-tumoral hepatic lineage (HL-7702). This finding was attributed to the pro-apoptotic mechanisms of hydroxytyrosol, namely, the induction of cell cycle arrest in the G2/M phase, increased cleavage by pro-caspase-3 of poly (ADP-ribose) polymerase (PARP), and the suppression of the PI3K/AKT pathway and nuclear factor kappa B (NF-κB) [52].

Currently, a study is investigating the action of hydroxytyrosol on cell viability and intracellular calcium levels ([Ca^2+^]i) in HepG2 hepatoma cells. Hydroxytyrosol at concentrations of 40 to 100 μM was found to inhibit cell viability in HepG2 hepatoma cells. In addition, hydroxytyrosol-induced increases in intracellular calcium levels ([Ca^2+^]i) have been observed, indicating a role for calcium signaling in the cytotoxic effects of the compound. The involvement of protein kinase C (PKC)-sensitive, store-operated calcium entry in the effects of hydroxytyrosol suggests a regulatory role of PKC in mediating the cellular responses to the compound. This study indicated that hydroxytyrosol-induced calcium release from thapsigargin-sensitive endoplasmic reticulum (ER) was through a phospholipase C (PLC)-dissociated pathway, highlighting alternative mechanisms by which the compound influences cell death processes. Thus, it indicated yet another route through which hydroxytyrosol could exert antitumor activity on hepatocellular carcinoma cells [53].

A preliminary study specifically investigated the impact of HT on HL60 cells, a type of human leukemia cell line, and its mechanism of action in inducing cell death. HT (50 to 100 μM) can completely stop the proliferation of HL60 cells and trigger apoptosis. This effect was not observed with a similar compound, tyrosol, which lacks the specific structural arrangement of hydroxyl groups found in HT. The study used flow cytometry and the detection of specific cellular markers, like poly (ADP-ribose) polymerase cleavage and caspase 3 activation, to confirm the apoptotic process. The apoptotic effect of HT is linked to its ability to prompt the early release of cytochrome c from the mitochondria, a key event in the intrinsic pathway of apoptosis. This release precedes the activation of caspase 8, ruling out the involvement of cell death receptors in the process. The compound was effective not only in HL60 cells but also in quiescent and differentiated HL60 cells, as well as in peripheral blood lymphocytes, indicating a broad impact on different cell types [45].

The mechanisms by which hydroxytyrosol promotes the induction of apoptosis in human leukemia cells were further investigated. The treatment of HL-60 cells with HT exerted a significant inhibitory effect on DNA synthesis, leading to a substantial reduction in cell proliferation. This is evidenced by a 92% decrease in the incorporation of [3H]-thymidine, a marker of DNA replication, at a concentration of 100 mmol/L. Moreover, the compound induced apoptosis, as demonstrated by the release of cytosolic nucleosomes and flow cytometry analysis. HT also influenced the cell cycle progression of HL60 cells by causing an accumulation in the G0/G1 phase, a resting phase before DNA replication, after 25 h of treatment. This arrest in the cell cycle is a key mechanism through which the compound inhibits cell growth and division. Furthermore, the compound reduced levels of cyclin-dependent kinase 6 (CDK6), a protein that promotes cell cycle progression, while increasing levels of cyclin D3, which is involved in cell cycle arrest. Additionally, it upregulated the expression of CDK inhibitors p21WAF1/Cip1 and p27Kip1, which are known to halt the cell cycle at the G0/G1 phase [46].

Parra-Perez et al. (2022) conducted cytotoxicity tests on three cell lines: Jurkat, HL60 (human leukemia T cells), and Raw264.7 (murine macrophages). They demonstrated that HT exhibited varying levels of toxicity, with IC_50_ values of 27.3 µg/mL, 109.8 µg/mL, and 45.7 µg/mL, respectively, at 24 h. This indicates that HT can selectively target cancer cells while being less toxic to normal cells. Furthermore, HT induced cell cycle arrest in the G0/G1 phase and increased apoptosis in Jurkat and HL60 cells, demonstrating its antiproliferative effects. These actions were associated with the inhibition of the PI3K signaling pathway and the activation of the MAPK pathway, both of which play crucial roles in cell growth and survival. In addition to its anticancer effects, HT showed anti-inflammatory properties by reducing levels of nitric oxide (NO) in Raw264.7 cells, which were previously stimulated to induce inflammation. This anti-inflammatory action was supported by changes in the expression of key markers of inflammation and cancer, further highlighting the potential of HT as a dual-action therapeutic agent. This evidence underscores the significant potential of hydroxytyrosol towards advances in the treatment of hematological neoplasms, particularly acute human leukemia, and as an anti-inflammatory agent [47].

Regarding in vitro studies with breast cancer cells, hydroxytyrosol has been explored less as a therapeutic alternative compared with oleuropein. Han et al. (2009) evaluated the effects of hydroxytyrosol and oleuropein in isolation. They showed that hydroxytyrosol (50 µg/mL) reduced cell proliferation and increased cell death (apoptosis) in MCF-7 cells. Moreover, hydroxytyrosol induced a block in the cell cycle of MCF-7 cells, specifically at the G1 to S phase transition, leading to an accumulation of cells in the G0/G1 phase, characteristic of apoptosis. The authors also highlighted the antioxidant properties of hydroxytyrosol, which reduce oxidative stress and DNA damage, helping prevent the uncontrolled growth of cancer cells [30].

The synergistic effects of hydroxytyrosol and squalene (SQ) on highly metastatic human breast tumor cells (MDA-MB-231) were investigated. When combined, HT at a concentration of 100 µM, along with varying levels of SQ, demonstrated significant antitumor activity. This combination effectively reduced the cell viability, promoted apoptosis, and induced DNA damage specifically in metastatic breast cancer cells. These actions collectively contribute to the suppression of cancer cell viability and proliferation, highlighting the potential of HT and SQ as chemopreventive agents [55].

Costantini et al. (2020) investigated the anti-cancer properties of hydroxytyrosol specifically in the context of metastatic melanoma (A375, HT-144, and M74 cells), known for its aggressiveness and resistance to traditional treatments, whose incidence is on the rise, particularly in Western populations. Hydroxytyrosol treatment (50 to 250 μM) led to a decrease in cell viability and an increase in apoptotic cell death, and upregulated the expression of pro-apoptotic proteins like p53 and γH2AX, known for their roles in the DNA damage response and cell death, while downregulating the expression of AKT, a protein that promotes cell survival and proliferation. Hydroxytyrosol treatment also inhibited the ability of melanoma cells to form colonies. The authors proposed that the observed effects of hydroxytyrosol on melanoma cells could be attributed to its ability to increase intracellular levels of ROS, leading to oxidative stress and DNA damage, which in turn trigger apoptosis and inhibit melanoma cell growth [54].

In summary, hydroxytyrosol demonstrates cytotoxic, antiproliferative, anti-inflammatory, and antioxidant properties in various cancer cell lines (Figure 3). Its ability to induce cell cycle arrest, regulate apoptosis, and inhibit key signaling pathways makes it a promising candidate for cancer therapy and warrants further investigation for its potential antitumor effects.

## 4. Anticancer Properties of Olive Leaf Extract

The olive leaf presents a complex plant matrix and contains strong antioxidants that may have chemopreventive properties [56]. The plant matrix plays an important role in the biological potential of these compounds by allowing synergistic and additive effects, different from those of the isolated compounds themselves [52,57]. The in vitro effects of olive leaf extracts (OLE, see Table 2) were investigated using the following cellular models: breast cancer, melanoma, cervix, ovarian, colon, colorectal adenocarcinoma, hepatocellular carcinoma, prostate, lung cancer, mesothelioma, pancreatic cancer, glioblastoma, neuroblastoma, and leukemias.

Nine studies investigated the antitumor effect of OLE on breast cancer cells. Initially, a study published in 2009 evaluated and standardized a method for its preparation. The authors performed two types of extraction: aqueous olive leaf dry extract (AOLE) and methanol olive leaf extract (MOLE). The content of phenolic compounds in both extracts was quantified, and the main phenolic compound found in both was oleuropein. The study investigated the extracts’ potential to suppress the proliferation of human breast adenocarcinoma (MCF-7) and human urinary bladder carcinoma (T-24) cells. Both extracts were reported to suppress cell proliferation of MCF-7 and T-24 cells at IC_50_ values of 209 and 178 μg/mL for AOLE and 174 and 510 μg/mL for MOLE [58].

Bouallagui et al. (2011) showed that treating MCF-7 breast cancer cells with hydroxytyrosol-rich OLE (2000 to 3000 μg/mL), extracted with methanol and water (4:1 *v*/*v*), significantly reduced the cell viability in a dose-dependent manner. This suppression was linked to the extract’s capacity to stop the cell cycle in the G0/G1 phase, which is an important checkpoint for cell division. Further molecular investigation revealed that the extract reduced the expression of two proteins: Pin1, which regulates the cell cycle, and cyclin D1, which stimulates cell division. Furthermore, the extract increased the expression of c-jun, an AP1 transcription factor linked with cell differentiation and apoptosis. These findings indicate that the OLE’s antitumoral action in MCF-7 cells is mediated by its effects on key proteins involved in cell cycle progression and cell division [59].

Regarding antitumor effects on breast cancer cells, a study published in 2013 investigated an OLE marketed as a nutraceutical in Spain. This study focused on the SKBR3 breast cancer cell line known for its resistance to certain treatments and also revealed that the most abundant compounds in this OLE were luteolin-7-*O*-glucoside, apigenin, and verbascoside. SKBR3 cells were exposed to the extract, and their uptake and metabolism of the phenolic compounds were monitored at different time points. Although oleuropein is the predominant compound in the extract, it was not found in the cells after incubation. Instead, compounds like luteolin and apigenin were identified within the cell cytoplasm, suggesting they could be the primary contributors to the observed cytotoxic effects on the SKBR3 cells [60]. Another study noted that OLE metabolites were found in breast cancer cells (JIMT-1 cells) after treatment (7.00 to 70.0 μg/mL). The metabolites found were apigenin, luteolin, and diosmetin. They were effective in inhibiting cancer cell growth and inducing apoptosis. This effect was attributed to the extract’s ability to inhibit the MAPK proliferation pathway, particularly at the extracellular signal-related kinase (ERK1/2) level [62].

The ability of OLE to act as an adjuvant in pharmacological treatment was evaluated in breast cancer cells of two lineages, MDA-MB-231 and MCF-7, known for their different responses to treatments. The OLE resulted in cytotoxic effects in a dose-dependent manner in both cell lines. The extract increased the cytotoxic effect of epirubicin in the MDA-MB-231 cell line. However, it blocked the cytotoxic effect of epirubicin in the MCF-7 cell line. The adjuvant effect of the drug appeared selective [63].

Oleuropein being a major component, particularly its demonstrated impact on triple-negative breast cancer (MDA-MB-231 cells) and ovarian cancer (OVCAR-3 cells), resulted in an IC_50_ close to 200 μg/mL for both cells, affecting cell viability, proliferation, apoptosis, and oxidative stress. OLE significantly inhibited the growth of both types of cancer cells, with a notable increase in apoptotic cell death. This was accompanied by a decrease in mitochondrial function and an increase in ROS production, indicating that OLE’s mechanism of action involves inducing oxidative stress within the cancer cells. Importantly, these effects were limited to cancer cells, leaving healthy cells unaffected [64].

Rashidipour and Heydari (2014) demonstrated that in the MCF-7 cell line, an OLE incorporated with silver nanoparticles (AgNPs) had an inhibitory concentration (IC_50_) at 24 h of incubation that was significantly lower (0.024 μg/mL) compared with the synthesized nanoparticles alone (50 μg/mL) [61]. Similar effects were demonstrated by Alhajri et al. (2022). In this study, OLEs synthesized Silver-Functionalized Carbon Nanotubes as a green approach (SFMWCNTs). The IC_50_ values for MCF7 cells treated with SFMWCNTs ranged from 15.78 to 375.10 μM at different time points (24, 48, and 72 h). HepG2 cells showed higher sensitivity with IC_50_ values of 69.49, 54.27, and 1.85 μM. Interestingly, the most cytotoxic activity was noticed against SW620 cells (human colorectal cancer) with lower IC_50_ values, which were 5.80, 4.97, and 0.49 μM for 24 h, 48 h, and 72 h, respectively. This suggests that the OLE not only aids in the synthesis of silver nanoparticles but also enhances their anticancer properties, and the effect is not restricted just to breast cancer cells [65].

OLE led to a significant decrease in the viability of HeLa cells, accompanied by the negative regulation of Cyclin-D1, a protein crucial for cell cycle progression, and the positive regulation of p21, a key player in promoting cell cycle arrest and apoptosis. These molecular changes indicate that OLE triggers intrinsic apoptosis in HeLa cells. In addition, the study showed the impact of OLE on NFkB, a transcription factor known for its role in promoting cancer progression. OLE reduced nuclear translocation, thus hampering its ability to activate genes that support the survival and proliferation of cancer cells. This effect is important in the context of cervical cancer, where the constitutive activation of NFkB, often stimulated by HPV oncoproteins, contributes to the progression of the disease. In addition, OLE neutralized epithelial-mesenchymal transition (EMT), a process involved in cancer metastasis, and inhibited independent and anchor-dependent cell growth induced by the epidermal growth factor. (EGF). These observations underline the wide-spectrum antitumor potential of OLE in HeLa cells. Additionally, the synergistic effect of OLE and cisplatin in reducing the viability of HeLa cells was attributed to the OLE’s ability to inhibit the NFkB, Akt, and MAPK pathways, all known to contribute to chemoresistance to cisplatin. By overcoming these resistance mechanisms, OLE increases the effectiveness of cisplatin, offering a potential reduction of chemoresistance in the treatment of cervical cancer [66].

Other studies also revealed an antitumor effect on hepatocellular carcinoma and colorectal cancer cells after treatment with OLE, in these cases using the crude extract. Regarding hepatocellular carcinoma cancer (HCC), Bektay, Güler, Gökçe, and Kızıltaş (2021) conducted their study using two types of cells: H4IIE *Rattus norvegicus* hepatoma cells (representing HCC) and *Rattus norvegicus* healthy liver clone-9 cells, treated with varying concentrations of OLE (250 to 2000 ppm). The OLE induced significant apoptotic, genotoxic, cytotoxic, and oxidative effects in H4IIE cells in a dose-dependent manner. These effects were significantly higher compared with the control groups, indicating OLE’s potential to selectively target and damage HCC cells [67].

Among the studies involving OLE and colorectal cancer, a study withHCT-116 colorectal cells assessed the cytotoxicity of the OLE from the olive cultivar Frantoio, which was the richest in phenols and triterpenoids, according to the characterization carried out in the study. The results revealed that the extract, rich in both phenols and triterpenoids, exhibited the most significant reduction in cell viability, with an IC_50_ value of 88.25 µg/mL. Importantly, a dose-dependent relationship was observed for all tested extracts (Frantoio, Leccino, and Moraiolo olive cultivars), suggesting a promising avenue for further research into the development of olive-based treatments for colorectal cancer [68].

Albogami and Hassan (2021) analyzed the effects of an AOLE, particularly with a high content of chlorogenic acid, using colorectal cancer cells (HT 29) and human prostate cells (PC3). The results indicated that HT29 cells had greater sensitivity to treatment. The IC_50_ values for PC3 cells after treatment were 553.8, 328.8, 236.6, and 203.9 μg/mL at 12, 24, 48, and 72 h, respectively. For HT29 cells after treatment, the IC_50_ values were 535.3, 289.6, 203.1, and 198.6 μg/mL in 12, 24, 48, and 72 h, respectively. These values were significantly lower for HT29 cells than for PC3 cells in 12 h (*p* < 0.05), 24 h (*p* < 0.001), and 48 h (*p* < 0.001), and were used in the investigation of the extract’s pathways of action. In general, extract not only killed cancer cells but also stopped their growth and changed their physical structure, leading to apoptosis. By observing DNA fragmentation and changes in gene activity related to cell death, the study confirmed that the extract’s main mechanism of action was through inducing apoptosis. The extract also increased stress within the cancer cells by raising their levels of ROS and reducing their antioxidant defenses. This imbalance makes it harder for cancer cells to survive [69].

The effect of OLE was also evaluated in terms of its radiomodulator potential. Once again thinking of OLE as a source of compounds with adjuvant effects on cancer treatment, in 2022, a study was published that carried out the characterization of an OLE and then evaluated its effects on two tumor cell lines and two non-tumor lines that also received X-ray irradiation treatment to mimic radiation therapy. DU145 prostate cancer cell lines and pancreatic PANC-1 cell lines were subjected to the same experimental conditions adopted for the non-cancer cell lines HUVEC and MCF-10A [70].

Among the identified compounds, oleacein stood out as the most abundant molecule in the extract. The cells were evaluated for their response in terms of micronucleus (MN) induction, a marker of DNA damage, and premature senescence (PS), a state of irreversible cell cycle arrest often induced by DNA damage. Pre-treatment with the olive leaf extract significantly reduced the frequency of radiation-induced MN and delayed the onset of PS in normal cells, indicating a protective effect against radiation-induced damage. In contrast, the extract exacerbated the genotoxic effects of ionizing radiation in cancer cells, leading to increased MN formation and accelerated PS. The extract’s ability to protect normal cells while making cancer cells more sensitive to radiation suggests it could be a valuable addition to radiotherapy, offering a way to target cancer more effectively while minimizing damage to healthy tissues [70].

The radiomodulator potential was an important discovery for investigating the antitumor potential of OLE in colorectal cancer cells and for pancreatic cancers. Until then, there was only one preliminary study that used MiaPaCa-2 pancreatic cancer cells to evaluate the viability of these cells through OLE treatment of Corregiola and Frantoio varieties (100 and 200 μg/mL) to compare the effects of OLE obtained by extraction methods with water, 50% ethanol, and 50% methanol, where the aqueous extract proved to be more efficient [71].

Two studies in 2011 assessed the antitumoral potential of OLEs in human leukemia cells (HL60 and Jurkat). The first study [72] carried out the characterization of OLE and extracted the main phenolic compounds identified for the treatment of human promyelocytic leukemia cells HL60 not only with OLE but also with their identified major compound, oleuropein, and luteolin (10 μL/mL, 170 μM, and 40 μM). OLE, oleuropein, and luteolin showed dose-dependent cytotoxicity with different IC_50_ values (10 μL/mL, 170 μM, and 40 μM, respectively). DNA fragmentation patterns and cell staining with acridine orange and ethidium bromide indicated that the mechanism for the cytotoxic effect of OLE, oleuropein, and luteolin was the apoptotic pathway, with DNA laddering and cytoplasmic and nuclear changes. To understand how the extracts affected cells, a second study [73] that year used Jurkat cells from human leukemia. The WST-1 proliferation kit and the [3H]-thymidine incorporation method confirmed the antiproliferative effect. In addition, they investigated whether cells were dying from apoptosis; coloring cells with Annexin V-FITC and PI (propidium iodide) and examining the expression of the main proteins involved in apoptosis (Bcl-2, Bax, and p53) revealed that olive leaf extracts were inducing apoptosis in leukemic cells.

Following that, a study employed K562 cells, a kind of leukemia cell recognized for its multipotency (ability to differentiate into several cell types), to explore the effect of Chemlali Olive Leaf Extract (COLE). The authors discovered that COLE treatment significantly reduced cell proliferation and caused cells to stop at many stages of the cell cycle, especially G0/G1 and G2/M. This cell cycle arrest is a critical process that stops cancer cell development and allows the cells to undergo apoptosis or differentiation. Importantly, the study discovered that COLE not only caused apoptosis but also induced the differentiation of K562 cells into monocytes, a kind of white blood cell engaged in the immune system. To understand the molecular mechanisms behind this differentiation-inducing effect, the researchers conducted a microarray analysis, which revealed the differential expression of several genes, including *IFI16*, *EGR1*, *NFYA*, *FOXP1*, *CXCL2*, *CXCL3*, and *CXCL8*. These genes are known to be involved in the differentiation of cells into the monocyte/macrophage lineage, providing strong evidence of the commitment of K562 cells to this specific differentiation pathway under the influence of COLE. To further understand the molecular processes behind this differentiation-inducing action, a microarray analysis was used that indicated the differential expression of many genes, including *IFI16*, *EGR1*, *NFYA*, *FOXP1*, *CXCL2*, *CXCL3*, and *CXCL8*. These genes are known to be involved in cell differentiation into the monocyte/macrophage lineage, indicating that K562 cells are committed to this differentiation pathway when exposed to COLE. These findings give insights into the mechanism by which olive leaf exerts its antileukemia action [74].

Malignant mesothelioma is a type of cancer that arises from the thin layer of tissue that covers the internal organs, most commonly the lungs and chest wall. It is notoriously difficult to treat, with a poor prognosis, particularly in advanced stages, as well as lung cancer. An OLE and an OLE enriched in oleuropein were analyzed, and both significantly reduced the viability of mesothelioma cells (REN cells) (IC_50_: 22 μg/mL). Changes in intracellular calcium levels ([Ca^2+^]) and the cells’ viability were investigated. By using fura-2 microspectrofluorimetry, a technique that allows the measurement of calcium levels within cells, the authors observed that the oleuropein-enriched fraction led to dose-dependent increases in cytosolic calcium concentrations. This effect was attributed to oleuropein’s interaction with T-type calcium channels, a type of channel known to be involved in cancer cell proliferation [75].

The major flavonoid compound of an OLE, morin, and OLE itself significantly reduced the growth of H460 lung cancer cells and increased cell death by apoptosis. The most notable effects were observed at concentrations of 800 μM for morin and 800 μg/mL for OLE, with morin showing greater effectiveness in the induction of apoptosis. The study also investigated the mechanism behind this cell death, revealing that both substances led to changes in the potential of the mitochondrial membrane, a fundamental step in the intrinsic pathway of apoptosis. This suggests that morin and OLE trigger a specific, controlled process of lung cancer cell death in H460 [76].

Glioblastoma multiforme (GBM) is one of the most aggressive and fatal types of brain tumors, necessitating new treatment options. OLE (2 mg/mL and 1 mg/mL), both independently and in combination with TMZ (Temozolomide), exhibited anti-proliferative effects on human glioblastoma cell lines (T98G). Furthermore, when cells were treated with both OLE and TMZ, there was a notable change in the expression levels of specific miRNAs, particularly those involved in cell cycle regulation and apoptosis. Specifically, *miR-181b*, *miR-153*, *miR-145*, *miR-137*, and *let-7d* were significantly upregulated after treatment with both TMZ and OLE. These changes suggest a potential synergistic effect between OLE and TMZ in targeting GB [77]. The same study group, using the same cell lineage and treatment with OLE and bevacizumab, assessed the impact on tumor weight, vascularization (the formation of blood vessels inside the tumor), invasiveness (the ability of cancer cells to spread in the surrounding tissue), and migration (the movement of cancerous cells). OLE at a concentration of 2 mg/mL led to a significant reduction in all these aspects (*p* = 0.0001 for tumor weight, *p* < 0.001 for vascularization, and *p* = 0.004 for invasiveness and migration). These effects were associated with a decrease in the expression of key proteins involved in angiogenesis (VEGFA) and invasion (MMP-2 and MMP-9). This suggests that OLE not only has autonomous potential but can also increase the effectiveness of existing treatments [78].

Neuroblastoma (NB) is the most prevalent solid tumor outside the brain in children. In a 2021 study, OLE significantly reduced the viability of NB cells in a dose- and time-dependent manner, both in traditional 2D cell cultures and more complex 3D models. This reduction was accompanied by a notable inhibition of cell proliferation, characterized by a halt in the cell cycle at the G0/G1 phase and an increase in cells undergoing apoptotic death, as evidenced by the accumulation of cells in the sub-G0 phase and the upregulation of key apoptotic proteins, such as caspases 3 and 7. Additionally, OLE demonstrated the ability to impede the migration of NB cells, a crucial aspect of cancer progression. Of particular interest was the synergistic effect observed when OLE was combined with the chemotherapy drug topotecan. This combination significantly enhanced the reduction in NB cell viability, suggesting a potential strategy for improving the efficacy of existing treatments [79].

De Cicco et al. (2022) observed that OLE inhibits the growth of melanoma cells, causing them to halt in their cell cycle (the cell division process) and inducing apoptosis. OLE also showed the ability to reduce the migration and invasion of melanoma cells, as well as its ability to form colonies. These effects were associated with a decrease in the expression of EMT-associated factors, suggesting that OLE could help prevent the transformation of melanoma cells into a more aggressive metastatic form [80].

An OLE enriched with oleuropein, OLEO, significantly reduced the glycolysis rate in A375 cells of human melanoma, a type of skin cancer, without affecting its oxidative phosphorylation, a metabolic pathway that is less favorable to cancer cells due to its slower energy production. This reduction in glycolysis has been associated with a decrease in the expression of three crucial proteins: glucose-1 transporter (GLUT1), protein kinase M2 isoform (PKM2), and monocarboxylate-4 transporter (MCT4), all of which play roles in facilitating the glycolytic process in cancer cells. This study extended its analyses to other types of cancer, including colon carcinoma, breast cancer, and chronic myeloid leukemia, and found that the metabolic effects of OLEO were not limited to melanoma cells. This broad-spectrum inhibition of glycolysis in different tumor cells highlights the antitumoral potential of OLEO, as previously seen [25].

In summary, OLE demonstrates cytotoxic, pro-apoptotic, anti-inflammatory, and antioxidant properties in various cancer cell lines. Its ability to induce cell cycle arrest, inhibit tumor growth, and modulate signaling pathways highlights its potential as a natural therapeutic agent for cancer treatment. Additionally, it also has chemical sensitizing and radiomodulating potential, see Figure 4.

## 5. Potential Antitumor Mechanisms of Oleuropein, Hydroxytyrosol, and Olive Leaf Extracts

With the evidence summarized in this review, we explored the numerous anticancer activities of oleuropein, hydroxytyrosol, and OLE as well as their molecular processes, according to Figure 5.

Oleuropein has shown promising anticancer effects: (1) Induction of apoptosis: Oleuropein has been observed to induce apoptosis in cancer cells, leading to cell death; (2) Cytotoxicity: Oleuropein exhibits cytotoxic effects on various cancer cell lines, including breast, melanoma, colon, cervix, and ovarian cancer cells; (3) Inhibition of cell proliferation: Oleuropein has been shown to inhibit cell proliferation in cancer cells, leading to a reduction in tumor growth; (4) Cell cycle arrest: High doses of oleuropein can induce cell cycle arrest in cancer cells, affecting different phases of the cell cycle; (5) Antioxidant and iron chelator: At lower doses, oleuropein acts as an antioxidant and iron chelator, protecting cells from oxidative stress-induced damage; (6) Inhibition of NF-κB Activation: Oleuropein has been reported to inhibit NF-κB activation, suppressing the NF-κB signaling cascade in cancer cells; (7) Selective toxicity: Oleuropein and synthetic analogues of oleuropein have shown selective toxicity towards cancer cells while sparing normal cells; (8) Enhanced antitumor activity: Oleuropein analogues have displayed potent in vivo activity against melanoma, retarding tumor growth and stimulating antitumor immune responses [19,21,22,30,31,32,33,34,35,36,37,38,39].

Hydroxytyrosol also exhibits significant anticancer effects: (1) Cytotoxicity: Hydroxytyrosol has demonstrated cytotoxic effects on various cancer cell lines, including human leukemia cells (Jurkat, HL60), breast cancer cells (MCF-7), and colorectal cancer cells (LS180, SW620); (2) Selective toxicity: HT has shown varying levels of toxicity towards cancer cells while being less toxic to normal cells, indicating its potential as a selective anticancer agent; (3) Cell cycle arrest: HT induces cell cycle arrest in cancer cells, particularly in the G0/G1 phase, inhibiting cell proliferation and promoting apoptosis; (4) Anti-Inflammatory properties: HT exhibits anti-inflammatory properties by reducing the levels of inflammatory markers, such as nitric oxide, in stimulated cells; (5) Modulation of signaling pathways: HT affects key signaling pathways involved in cell growth and survival, including the PI3K and MAPK pathways, contributing to its antiproliferative effects; (6) Antioxidant activity: HT enhances the expression of antioxidant enzymes and reduces oxidative stress in cancer cells, potentially preventing cell damage and proliferation; (7) Regulation of apoptosis: HT influences the expression of genes involved in apoptosis, promoting cell death and inhibiting cancer cell survival; (8) Inhibition of TrxR1 activity: HT effectively inhibits TrxR1 activity, leading to the accumulation of ROS within cancer cells and inducing cell death [40,41,42,43,44,45,46,47,48,49,50,51,52,53,54,55].

Nevertheless, we must acknowledge that the evidence supporting the effects of the isolated compounds is limited due to the wide range of concentrations used for treatment; additionally, studies with oleuropein have primarily used breast cancer cell lines, whereas studies with HT have focused on colorectal and hepatic cells.

OLE contains a combination of beneficial compounds that contribute to its anticancer effects, such as: (1) Cytotoxicity: Olive leaf extract has demonstrated cytotoxic effects on various cancer cell lines, including breast, melanoma, cervix, ovarian, colon, and hepatocellular carcinoma cells; (2) Induction of apoptosis: OLE has been shown to induce apoptosis in cancer cells, leading to programmed cell death and the inhibition of tumor growth; (3) Cell cycle arrest: OLE can induce cell cycle arrest in cancer cells at different stages, preventing cell proliferation and promoting cell death; (4) Anti-inflammatory properties: OLE exhibits anti-inflammatory properties by reducing the levels of inflammatory markers in stimulated cells, potentially inhibiting cancer progression; (5) Antioxidant activity: OLE acts as an antioxidant, protecting cells from oxidative stress-induced damage and enhancing cellular defense mechanisms; (6) Inhibition of tumor growth: OLE has been reported to inhibit the ability of cancer cells to form colonies, thereby suppressing tumor growth and metastasis; (7) Regulation of signaling pathways: OLE modulates key signaling pathways involved in cell growth and survival, contributing to its antiproliferative effects; (8) Adjuvant therapy: OLE has been investigated as an adjuvant therapy in combination with standard cancer treatments, such as chemotherapy and radiotherapy, to improve treatment response and reduce side effects [25,56,57,58,59,60,61,62,63,64,65,66,67,68,69,70,71,72,73,74,75,76,77,78,79,80].

Regarding the effects of OLE in a combined drug or pharmacological treatment, the following were observed: (1) Enhanced antitumor effects: Studies have shown that OLE can enhance the antitumor effects of conventional cancer drugs by increasing their efficacy in inhibiting cancer cell growth and inducing apoptosis; (2) Radiosensitizing effects: OLE has been found to sensitize cancer cells to radiation therapy, making them more susceptible to the effects of ionizing radiation and enhancing the overall treatment outcomes; (3) Selective toxicity enhancement: When combined with pharmacological agents, OLE has demonstrated the ability to enhance the selective toxicity of drugs towards cancer cells while protecting normal cells; (4) Synergistic effects: OLE has shown synergistic effects when combined with certain drugs, leading to enhanced cytotoxicity, cell cycle arrest, and apoptosis in cancer cells; (5) Radiomodulator potential: OLE has been identified as a radiomodulator, enhancing the effects of radiotherapy on cancer cells while protecting normal cells from radiation-induced damage; (6) Chemopreventive properties: OLE’s antioxidant and anti-inflammatory properties may complement the effects of pharmacological agents in preventing cancer development and progression [63,66,70,78,79].

Overall, the combination of OLE with drugs or pharmacological treatments has shown promising results in enhancing the efficacy of cancer therapies, improving treatment outcomes, and reducing the toxic effects on normal cells. Further research is needed to explore the full potential of OLE as a complementary therapy in combination with conventional cancer treatments.

## 6. Final Considerations

This review, based on in vitro evidence, provides a detailed description and discussion of the mechanisms by which bioactive compounds from olive leaves might act against cancer cells and the potential of these compounds to increase the sensitivity of cancer cells to conventional anticancer therapy. OLE exhibits significant antitumor properties, likely due to its rich content of bioactive compounds. These compounds may interact synergistically to target multiple pathways involved in cancer development and progression. The study of the anticancer properties of OLE represents an important step in the search for natural and effective cancer therapies. This review highlighted the importance of using whole plant extracts rather than isolated compounds, as they have been shown to inhibit tumor cell growth and induce cell death. These findings enhance our scientific understanding of the bioactivity of olive leaves and pave the way for future research into the specific mechanisms underlying their anticancer potential. The findings suggest that olive leaves and their bioactive compounds have promising anticancer properties, making them potential candidates for further research on the development of cancer treatment strategies. Recent technologies delineate areas for new interactions between food engineering and the medical field to aid in the controlled delivery of cancer drugs, to develop models for the study of cancer, and to ensure more attractive and proper nutrition for cancer patients. By taking advantage of this technology, further possibilities for the development of new applications of OLE can be considered.

## Figures and Tables

**Figure 1 molecules-29-04249-f001:**
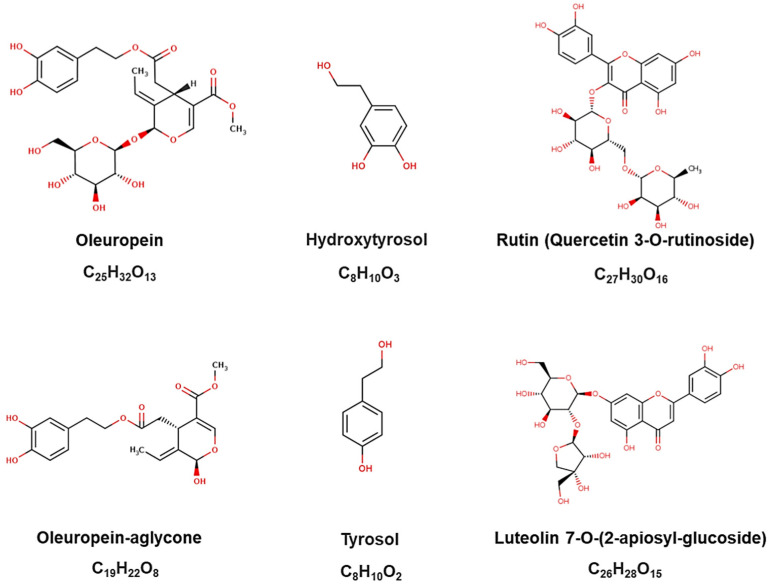
Chemical structures of the major phenolic compounds identified in olive leaves, as provided by the Phenol-Explorer platform.

**Figure 2 molecules-29-04249-f002:**
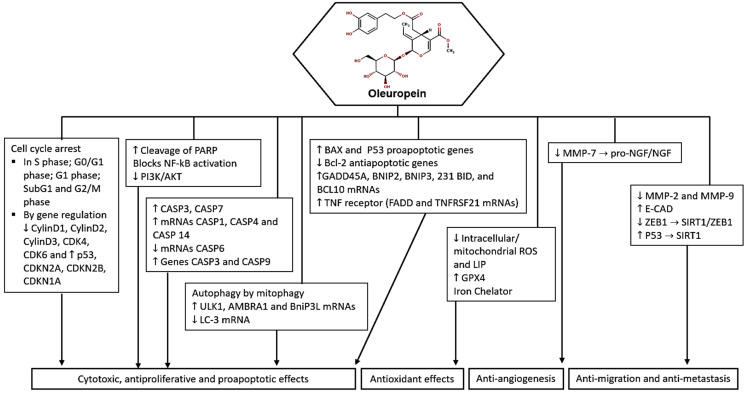
Potential molecular targets and signaling pathways for the anticancer effect of oleuropein. ↑—Upregulation capacity or enhanced effect; ↓—Downregulation capacity or reducing effect.

**Figure 3 molecules-29-04249-f003:**
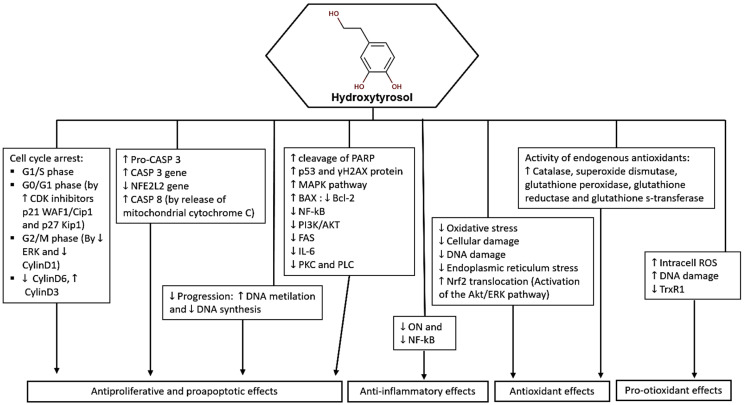
Potential molecular targets and signaling pathways for the anticancer effect of HT. ↑—Upregulation capacity or enhanced effect; ↓—Downregulation capacity or reducing effect.

**Figure 4 molecules-29-04249-f004:**
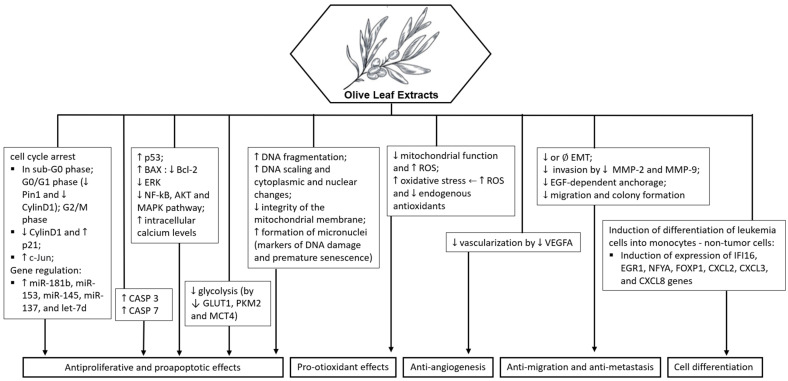
Potential molecular targets and signaling pathways for the anticancer effect of olive leaf extract. ↑—Upregulation capacity or enhanced effect; ↓—Downregulation capacity or reducing effect.

**Figure 5 molecules-29-04249-f005:**
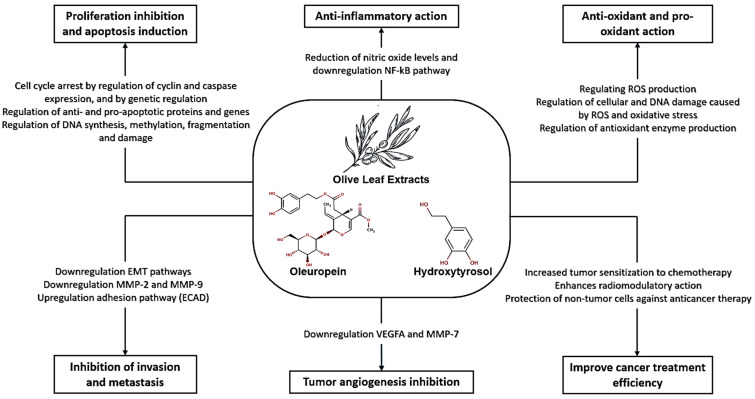
Mechanisms of oleuropein, hydroxytyrosol, and olive leaf extracts against cancer.

**Table 1 molecules-29-04249-t001:** A summary of studies based on cell lines investigating the anticancer effects of the main compounds found in olive leaves.

Compound	Concentration Range	Types	Cell Models	Observed Effects	Refs.
Oleuropein and Hydroxytyrosol	100 and 200 μg/mL	Breast	MCF-7	↓ Cell viability; ↑ morphological changes indicative of apoptosis; ↑ caspases and cell cycle arrest at G1 phase.	[30]
Oleuropein	1, 10, and 100 μM	Breast	T-47D and MCF-7	↓ Cell viability and cell cycle arrest at the G2/M phase.	[21]
Oleuropein	100 or 200 μM	Breast	MCF-7	↑ *Bax* gene and ↑ activation of *p53*-dependent apoptotic pathways, by ↑ *p53* gene expression.	[31]
Oleuropein and doxorubicin (DOX)	300 or 600 µg/mL and 0.05 to 1 µM	Breast	MCF-7	Oleuropein: ↓ cell migration and proliferation, ↓ MMP-2/9, ↓ ZEB1, ↑ E-CAD and p53; Oleuropein-DOX: ↑ antiproliferative effect, ↓ DOX IC_50_ by 6.40 times, ↑ ratio of late/early apoptosis.	[32]
Oleuropein	0 to 100 μM	Breast	MDA-MB-231 and MCF-7	↓ Cell viability and migration; cell cycle arrest at the sub-G1 phase; ↑ apoptosis with ↑ cleavage of PARP and caspase-3/7. ↓ NF-κB activation.	[33]
Oleuropein	0 to 700 µM	Breast	MDA-MB-231 and MDA-MB-468	↓ Cell viability; apoptosis was mainly induced by the ↓ antiapoptotic gene *TNFRSF11B* and *BIRC5* and ↑ *CASP4*.	[34]
Semi-synthetic oleuropein analogs	15 and 20 μM	Breast, melanoma, cervix, and colon	MCF-7 and SKBR3, FM3, HeLa, and HCT-116	↑ Cancer cell death without causing severe toxicity in non-tumor cells.	[22]
Oleuropein	100, 200 and 400 µM	Ovarian and breast	HEY and MCF-7	Antioxidant activity (iron chelator; ↑ ROS scavenging enzyme GPX4; ↓ LIP and mitochondrial ROS); cell cycle arrest at the subG1 and G2/M phases.	[19]
Oleuropein	20 to 100 μM	Hepatocellular carcinoma	HepG2	Morphological alterations; ↓ cell growth; ↑ caspase and Bcl-2 family and ↓ PI3K/AKT signaling; no effect on cell viability.	[35]
Oleuropein	10 to 100 µmol/L	Hepatocellular carcinoma	HepG2	Protective effects on cellular viability, accompanied by ↓ of Casp-3 cleavage.	[36]
Oleuropein and cisplatine	200 μM and 50 μM	Hepatocellular carcinoma	HepG2	Pro-NGF/NGF balance via affecting MMP-7 activity.	[37]
Oleuropein or Oleuropein + Adriamycin (ADR)	10 to 50 μg/mL	Osteosarcoma	MG-63	Cell cycle arrest at the G0/G1 phase; ↑ ADR cytotoxicity; ↑ *ULK1*, *AMBRA1*, and *BniP3L* mRNAs; ↑ LC-3.	[38]
Oleuropein	25 to 800 µmol/L	Neuroblastoma	SH-SY5Y	↓ Cell viability; ↓ genes involved in cell cycle progression; ↑ genes that promote cell cycle arrest.	[39]
Hydroxytyrosol and tyrosol	250 μmol/L and 500 μmol/L	Colorectal adenocarcinoma	Caco-2	HT ↓ cellular damage induced by oxidative stress	[40]
Hydroxytyrosol	5.0 to 162.5 μM	Colorectal adenocarcinoma	Caco-2	Cell cycle arrest at the G2/M phase; ↓ cell division and proliferation; ↓ ERK and cyclin D1 expression.	[41]
Hydroxytyrosol	50, 100 and 150 μM	Colorectal adenocarcinoma	LS180	↑ *CASP3* gene, ↑ cell death by BAX:BCL2 ratio; ↓ *NFE2L2* gene; ↑ activity of antioxidant enzymes.	[42]
Hydroxytyrosol	5, 10 and 20 μM	Colorectal adenocarcinoma	SW620	↑ Cell death by ↑ TrxR1 activity and leads to the cell accumulation of ROS; ↑ apoptosis and G1/S cell cycle arrest.	[43]
Hydroxytyrosol	0, 5, 10, 50, 100 or 150 µM	Colorectal adenocarcinoma	Caco-2	↓ Cancer development and progression by ↑ DNA methylation, the *EDNRA* gene stood out as a target.	[44]
Hydroxytyrosol	50 to 100 μM	Myeloid leukemia and Colorectal cancer cells	HL60, HT-29 and CaCo2	HL60: ↑ apoptosis by release of cytochrome c from the mitochondria. HT-29 and CaCo2: HT was not able to induce cell death.	[45]
Hydroxytyrosol	100 μmol/L	Myeloid leukemia	HL60	↓ Cell proliferation; ↓ incorporation of [3H]-thymidine; ↑ release of cytosolic nucleosomes; cell cycle arrest at the G0/G1 phase; ↓ CDK6 and ↑ cyclin D3; ↑ CDK inhibi-tors p21WAF1/Cip1 and p27Kip1.	[46]
Hydroxytyrosol	6.25 to 50 μg/mL	Liquid cancers, leukemias, myelomas, and lymphomas	Jurkat, HL60 and Raw264.7	Cycle arrest in the G0/G1 phase and apoptosis, while being less toxic to normal cells; ↓ PI3K signaling pathway and ↑ MAPK pathway; ↓ NO	[47]
Hydroxytyrosol	10 to 40 μM	Hepatocellular carcinoma	HepG2	No changes in cell integrity or antioxidant status were verified.	[48]
Hydroxytyrosol	0.5, 1.0, 5.0 and 10.0 μM	Hepatocellular carcinoma	HepG2	↑ Expression of antioxidant enzymes; ↑ activation of AKT and ERK; ↑ nuclear translocation of the Nrf2 transcription factor.	[49]
Hydroxytyrosol	30 to 200 μM	Hepatocellular carcinoma	Hep3B e HepG2	Antiproliferative effect by ↑ FAS; cellular antioxidant system; ↓ IL-6.	[50]
Hydroxytyrosol	1 μM and 5 μM	Hepatocellular carcinoma	HepG2	↓ Endoplasmic reticulum stress.	[51]
Hydroxytyrosol	100 to 400 μM	Hepatocellular carcinoma	HepG2, Hep3B, SK-HEP-1 and Huh-7	↓ Cell proliferation; cell cycle arrest in the G2/M phase; ↑ cleavage PARP; ↓ PI3K/AKT pathway and NF-κB.	[52]
Hydroxytyrosol	0 to 100 μM	Hepatocellular carcinoma	HepG2	↓ Cell viability; ↑ intracellular calcium levels.	[53]
Hydroxytyrosol	50 to 250 μM	Melanoma	A375, HT-144, and M74	↓ Cell viability; ↑ apoptotic; ↑ p53 and γH2AX; ↓ AKT; ↓ colony formation; ↑ oxidative stress and DNA damage.	[54]
Squalene and HT	0.01 μM to 100 μM	Breast	MDA-MB-231	↓ Cell proliferation; ↑ apoptosis; ↑ DNA damage.	[55]

↑—Upregulation capacity or enhanced effect; ↓—Downregulation capacity or reducing effect.

**Table 2 molecules-29-04249-t002:** A summary of studies based on cell lines about the anticancer effects of olive leaf extract (OLE).

Compound	Concentration Range	Types	Cell Models	Observed Effects	Refs.
OLE	66 to 510 μg/mL	Breast and urinary bladder	MCF-7	↓ Cell proliferation.	[58]
OLE enriched in HT	2000, 2200, 2400, 2600, 2800, and 3000 μg/mL	Breast	MCF-7	↓ Cell viability; cell cycle arrest at the G0/G1 phase; ↓ Pin1 and cyclin D1.	[59]
OLE	200 μg/mL	Breast	SKBR3	↑ Cytotoxic effects.	[60]
OLE	50 and 0.024 μg/mL	Breast	MCF-7	↓ Cell proliferation.	[61]
OLE	7.00 and 70.0 μg/mL	Breast	JIMT-1	↓ Cell growth; ↑ apoptosis; ↓ MAPK pathway.	[62]
OLE and epirubicin	3.12 to 400 µg/mL	Breast	MCF-7 and MDA-MB-231	↑ Cytotoxicity of the drug only in MDA-MB-231.	[63]
OLE	100 to 400 μg/mL	Breast and ovarian	MDA-MB-231 and OVCAR-3 cells	↓ Cell viability and proliferation; ↑ apoptosis; ↓ oxidative stress; leaving healthy cells unaffected.	[64]
OLE	100, 50, 25, 10 and 5 μg/mL	Breast	SKBr3, AMJ13, MDA-MB-231 and MCF-7	↑ Cytotoxic effects.	[65]
OLE and OLE and Cisplatin	5 mM	Cervical cancer	HeLa	↓ Cell viability and Cyclin-D1; ↑ p21; neutralized EMT; ↓ NFkB, Akt and MAPK pathways.	[66]
OLE	250, 500, 1000 and 2000 ppm	Hepatocellular carcinoma	H4IIE cells	↑ Apoptotic, genotoxic, cytotoxic, and oxidative effects.	[67]
Extract of Phenolic Compounds and Triterpenes of branch an leaves Olive	97.06 μg/mL	Colon cancer	HCT-116	↓ Cell viability.	[68]
OLE	535.3, 289.6, 203.1, and 198.6 µg/mL	Colorectal and prostate	HT29 and PC3	↑ Apoptosis; ↑ DNA fragmentation and ROS and ↓ antioxidant defenses.	[69]
OLE (pre-treatment) and Cell Irradiation	12.5 μg/mL	Prostate and pancreatic	MCF-10A and DU145 cells, compared with normal HUVECs and MCF-10A cells	↑ Genotoxic effects of radiation in cancer cells and protected normal cells.	[70]
OLE	0 to 200 µg/mL	Pancreatic cancer	MiaPaCa-2	↓ Cell viability.	[71]
OLE	10 μL/mL, 170 μM, and 40 μM	Leukemia	HL60	DNA fragmentation and laddering; cytoplasmic and nuclear changes.	[72]
OLE	0.025, 0.05, 0.1, 0.2, 0.4, 0.6, 0.9, 1.1 mg dw	Leukemia	Jurkat	↓ Cell proliferation; ↑ apoptosis; ↓ Bcl-2; ↑ Bax, and p53.	[73]
OLE	50, 100, and 150 μg/mL	Chronic myelogenous leukemia	K562 Cells	↓ Cell proliferation and apoptosis by cell cycle arrest G0/G1 and G2/M; ↑ cell differentiation in monocytes.	[74]
OLE enriched in oleuropein or Oleuropein or HT	10 to 100 μM	Mesothelioma	REN cell	↓ Cell viability; ↑ cytosolic calcium.	[75]
OLE and flavonoid morin	50, 100, 200, 400 and 800 μM e μg/mL	Lung cancer	H460	↓ Cell growth; ↑ apoptosis by changes in the mitochondrial membrane.	[76]
OLE and OLE and TMZ	1 to 2 mg/mL	Glioblastoma	T98G	↑ Antiproliferative effects; ↑ miR-181b, miR-153, miR-145, miR-137, and let-7d.	[77]
OLE and OLE and bevacizumab	1 to 2 mg/mL	Glioblastoma	T98G	↑ Angiogenesis (↓ VEGFA); ↑ invasion (↓ MMP-2/9).	[78]
OLE and combining OLE with the chemotherapeutic topotecan	50 to 300 μM	Neuroblastoma	HTLA-230, IMR-32, SH-SY5Y and SK-N-AS	↓ Cell proliferation; cell cycle arrest at the G0/G1 phase and sub-G0 phase; ↑ caspases 3 and 7.	[79]
OLE	50 to 100 a 150 to 200 µg/mL	Human melanoma	A375 cells	↑ Apoptosis; ↓ migration, invasion, and ability to form colonies; ↓ EMT-associated factors.	[80]
OLEO (olive leaf extract enriched in Oleuropein)	6.25 µM to 800 µM	Human melanoma	A375	↓ Glycolysis rate; ↓ GLUT1, PKM2, and MCT4.	[25]

↑ Upregulation capacity or enhanced effect; ↓—Downregulation capacity or reducing effect.

## Data Availability

Data are contained within the article.

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
