# Peer review of "Olive Leaves as a Source of Anticancer Compounds: In Vitro Evidence and Mechanisms"

_molecules, 2024, doi:10.3390/molecules29174249_

Round 1

Reviewer 1 Report

Comments and Suggestions for Authors

Dear authors of manuscript No. molecules-3062628: Olive leaves as a source of anticancer compounds: in vitro evidence and mechanisms.,

Overall, I can state that the submitted manuscript elaborates on an interesting topic and is conceptually well-prepared and thought out. The structure of the publication corresponds to the study's title and the text is balanced in scope and content. I believe that the manuscript presents an interesting and comprehensively developed topic and will suitably complement another review already published earlier. I have no major comments on the text. However, I present a few small recommendations below.

1. Table 1 - In the "Observee effects" column, I recommend always starting with a capital letter (unification within the table).

2. There is an inconsistent expression of concentration ranges across the tables - compare Tab 1 (e.g., 0 to 100 uM) and Tab 2 (e.g., 3.12-400 ug/mL). The need to unify throughout the text.

3. L16 - use a different and more appropriate wording to explain the arrows.

4. Figures 2, 3, Table 2, etc. - capital letters at the beginning of the given text/bullet (it is not even uniform within the object, nor across objects either).

5. Table 2 - the hyphen is not used uniformly and correctly within the object (different character sizes, typographically incorrect). The same appears throughout the text, e.g., L58, etc.

6. Manuscript contains 2 objects labeled Figure 3! Inappropriate referring to objects in the text also needs to be corrected.

Author Response

Rio de Janeiro, August 15th, 2024.

Response to Reviewer 1 Comments

“Olive leaves as a source of anticancer compounds: in vitro evidence and mechanisms” No. molecules-3062628

Thank you very much for taking the time to review this manuscript. Please find the detailed responses below and the corresponding revisions/corrections highlighted in red color in the re-submitted files.

On behalf of all authors, sincerely,

Prof. Danielly C. Ferraz da Costa, Ph.D.

Institute of Nutrition

Rio de Janeiro State University

São Francisco Xavier, 524, Pavilhão João Lyra Filho, 12º andar,

Sala 12.144, Bloco F, 20550-013, Rio de Janeiro, RJ – Brazil

Phone: +55 21 2334-1037

·         Point-by-point response to Comments and Suggestions for Authors

Comments 1: Table 1 - In the "Observee effects" column, I recommend always starting with a capital letter (unification within the table).

Response 1: Thank you for pointing this out. We have modified. (Table 1 - L163 - "Observee effects" column)

Comments 2: There is an inconsistent expression of concentration ranges across the tables - compare Tab 1 (e.g., 0 to 100 uM) and Tab 2 (e.g., 3.12-400 ug/mL). The need to unify throughout the text.

Response 2: Thank you for pointing this out. We have modified. (Table 1, table 2 and table 3 – “Concentration range” column).

Comments 3: L164 - use a different and more appropriate wording to explain the arrows.

Response 3: Thank you, we agree with the comment and we have changed “* ↑-up regulation or enhancing ↓-down regulation or reducing.” to “* ↑-Upregulation capacity or enhanced effect; ↓-Downregulation capacity or reducing effect“. (L164 and L557)

Comments 4: Figures 2, 3, Table 2, etc. - capital letters at the beginning of the given text/bullet (it is not even uniform within the object, nor across objects either).

Response 4: Thank you. We agree with the comment, and we made the adjustments. (Figure 2 – L364, Figure 3 – L543, Figure 4 – L800, Figure 5 – 877, Table – L555.)

Comments 5: Table 2 - the hyphen is not used uniformly and correctly within the object (different character sizes, typographically incorrect). The same appears throughout the text, e.g., L58, etc.

Response 5: Thank you. We have modified. (L198; L221; L234; L272; L458; L587; Figure 2 and Figure 3)

Comments 6: Manuscript contains 2 objects labeled Figure 3! Inappropriate referring to objects in the text also needs to be corrected.

Response 6: Thank you for pointing this out. We have modified. (L801 - figure 4, L807 - figure 5), L882 - figure 5).

The suggestions of the reviewer were adopted, and the manuscript was readjusted. We thank the reviewer for the suggestions, and we hope that now the revised manuscript corresponds to the reviewer’s expectations.

Yours sincerely,

Prof. Danielly C. Ferraz da Costa, Ph.D.

Reviewer 2 Report

Comments and Suggestions for Authors

The review is well represented and added by colorful illustrations. The article included 80 references, which indicates a large-scale work. The work is well structured by topic, making it easy to read. I think this article will be helpful for chemists in natural products fields as well as scientists in medicine. However, this article can be accepted to publish after some minor revision:
1) Figure 1 should be mentioned in the main text.
2) The structures are small and not clear, the ethylidene bond in oleuropein needs to be fixed.
3) Why did the authors consider rutin and luteolin in the extract, if there is no further data on their biological activity. Either the data should be presented or structures should be separated outside the figure so that the reader does not get the impression that all of the presented structures will be considered. Another solution is to move the figure to section 1, where the general composition of the extract is described, and in section 3 explain why rutin and luteolin are not considered in this review.
4) In addition, it would be more logical if oleuropein and its derivatives tyrosol in the figure will be given next to each other.
5) In the table 1 for Oleuropein + doxorubicin should be given a definition of DOX as by analogy was done for Oleuropein + Adriamycin (ADR)
6) HT should be deciphered where the abbreviation was first indicated, or where the data on hydroxytyrosol begins, provide in brackets the abbreviation that will be used)
7) Ref 22 in the table: if possible, please provide what analogs (what functional groups or something like that)
8) IC50 should be subscript. Please check everywhere

Comments on the Quality of English Language

Minor editing of English language required

Author Response

Rio de Janeiro, August 15th, 2024.

Response to Reviewer 2 Comments

“Olive leaves as a source of anticancer compounds: in vitro evidence and mechanisms” No. molecules-3062628

Thank you very much for taking the time to review this manuscript. Please find the detailed responses below and the corresponding revisions/corrections highlighted in red color in the re-submitted files.

On behalf of all authors, sincerely,

Prof. Danielly C. Ferraz da Costa, Ph.D.

Institute of Nutrition

Rio de Janeiro State University

São Francisco Xavier, 524, Pavilhão João Lyra Filho, 12º andar,

Sala 12.144, Bloco F, 20550-013, Rio de Janeiro, RJ – Brazil

Phone: +55 21 2334-1037

·         Point-by-point response to Comments and Suggestions for Authors

Comments 1: Figure 1 should be mentioned in the main text.

Response 1: Thank you for pointing this out. We mentioned Figure 1. (L47).

Comments 2: The structures are small and not clear, the ethylidene bond in oleuropein needs to be fixed.

Response 2: Thank you. We have modified. (L56).

Comments 3: Why did the authors consider rutin and luteolin in the extract, if there is no further data on their biological activity. Either the data should be presented or structures should be separated outside the figure so that the reader does not get the impression that all of the presented structures will be considered. Another solution is to move the figure to section 1, where the general composition of the extract is described, and in section 3 explain why rutin and luteolin are not considered in this review.

Response 3: Thanks for the comment; we agree and we amended according to the suggestion. Figure 1 is placed now in Section 1—L56, and mentioned in L44-47. (L44-47 and L56).

Comments 4: In addition, it would be more logical if oleuropein and its derivatives tyrosol in the figure will be given next to each other.

Response 4: Thank you for the comment. We modified the order in which the compounds appear in the figure. (L56).

Comments 5: In the table 1 for Oleuropein + doxorubicin should be given a definition of DOX as by analogy was done for Oleuropein + Adriamycin (ADR)

Response 5: Thank you for pointing this out. We have modified. (Table 1)

Comments 6: HT should be deciphered where the abbreviation was first indicated, or where the data on hydroxytyrosol begins, provide in brackets the abbreviation that will be used)

Response 6: Thanks for pointing this out. The first time the term appears, also appears its abbreviation (L46).

Comments 7: Ref 22 in the table: if possible, please provide what analogs (what functional groups or something like that).

Response 7: Thanks for the comment. We specified the analogue (24) and described the difference in its chemical structure. (L265; L269 – L271).

Comments 8: IC50 should be subscript. Please check everywhere.

Response 8: Thank you. We have modified. (Tab 1; L197; 199; 252; 256; 326; 346; 497; 565; 600; 607; 611; 613; 615; 650; 657; 658; 698; 733).

The suggestions of the reviewer were adopted, and the manuscript was readjusted. We thank the reviewer for the suggestions, and we hope that now the revised manuscript corresponds to the reviewer’s expectations.

Yours sincerely,

Prof. Danielly C. Ferraz da Costa, Ph.D.

Reviewer 3 Report

Comments and Suggestions for Authors

Pessoa et al. describe in this manuscript the anticancer activity of compounds isolated from olive oil leaves.

I accepted to review the manuscript with enthusiasm and interest. However, my excitement was dampened by the lack of consistency of the manuscript together with the evidence-limited in vitro studies ( supposedly there are no in vivo and this tells a lot). The concentrations used for the two major compounds oleuropein and hydroxytyrosol are quite high and likely of limited significance and this may explain the lack of in vivo evidence. The majority of studies are performed in the same cell lines (i.e. MCF-7, MDA-231)/ There is confusion about the words antiproliferative, cell viability, and apoptosis and the impression is that they are used as they are synonyms. Similarly, the effect of migration for compounds that affect cell viability is of limited value. It is very difficult to assess the claims of the manuscript since very often reviews are cited and not original articles. For instance: “ This is due to the synergy of all bioactive compounds present in the extracts, which most likely affect their absorption and bioavailability [3].

Reference 3 is a review. (References 3 and 25 are the same)

In addition, there is a lack of consistency with references sometimes using numbers other than the author's name (i.e. Hormozi et al., Bulotta et al., Samara et al. etc).

The abbreviation HT I used I guess for hydroxytyrosol; if this is the case the authors use HT and hydroxytyrosol throughout the manuscript making the reading quite annoying.

Comments on the Quality of English Language

minor editing

Author Response

Rio de Janeiro, August 15th, 2024.

Response to Reviewer 3 Comments

“Olive leaves as a source of anticancer compounds: in vitro evidence and mechanisms” No. molecules-3062628

Thank you very much for taking the time to review this manuscript. Please find the detailed responses below and the corresponding revisions/corrections highlighted in red color in the re-submitted files.

On behalf of all authors, sincerely,

Prof. Danielly C. Ferraz da Costa, Ph.D.

Institute of Nutrition

Rio de Janeiro State University

São Francisco Xavier, 524, Pavilhão João Lyra Filho, 12º andar,

Sala 12.144, Bloco F, 20550-013, Rio de Janeiro, RJ – Brazil

Phone: +55 21 2334-1037

·         Point-by-point response to Comments and Suggestions for Authors

Comments 1: The concentrations used for the two major compounds oleuropein and hydroxytyrosol are quite high and likely of limited significance and this may explain the lack of in vivo evidence. The majority of studies are performed in the same cell lines (i.e. MCF-7, MDA-231).

Response 1: Thanks for the comment; We have added a paragraph mentioning this limitation. (L840 – L843).

Comments 2: There is confusion about the words antiproliferative, cell viability, and apoptosis and the impression is that they are used as they are synonyms.

Response 2: Thank you for your observation. In order to enhance the understanding of the differences between the terms, we included the effect assessment method and adjusted the term based on what the method evaluates. (Table 1 and Table 2; L173; L180; L196; L325; L199; L329; L344; L523; L569; L650).

Comments 3: It is very difficult to assess the claims of the manuscript since very often reviews are cited and not original articles. For instance: “This is due to the synergy of all bioactive compounds present in the extracts, which most likely affect their absorption and bioavailability [3].

Response 3: Thanks for the comment. We checked the revisions referenced in the manuscript, and we believe that for the introduction they contribute with the synthesis of already consolidated evidence. In topic 2, we try to reduce the number of revisions quoted to build the presentation of the compounds and the composition of the extract. Since Ref 13 is a systematic review—a scientific publication that compiles and summarizes scientific evidence—it was retained.; Ref 18 (L98) was updated by the original article cited therein; Ref 3 was replaced by Ref 26 (original article) (L128); we maintained Ref 28, and added an original article, Ref 27, addressing the information cited (L139). (L98; L128; L139).

Comments 4: Reference 3 is a review. (References 3 and 25 are the same). In addition, there is a lack of consistency with references sometimes using numbers other than the author's name (i.e. Hormozi et al., Bulotta et al., Samara et al. etc).

Response 4: Thank you for your observation. We removed a duplicate reference. The following references were renumbered after we made necessary corrections. (L125).

Comments 5: The abbreviation HT I used I guess for hydroxytyrosol; if this is the case the authors use HT and hydroxytyrosol throughout the manuscript making the reading quite annoying.

Response 5: Thanks for pointed this out. The first time the term appears, also appears its abbreviation. (L46).

·         Response to Comments on the Quality of English Language

Point 1: “Minor editing of English language required.”

Response 1: Thanks for this recommendation. A specialist reviewed the English language.

The suggestions of the reviewer were adopted, and the manuscript was readjusted. We thank the reviewer for the suggestions, and we hope that now the revised manuscript corresponds to the reviewer’s expectations.

Yours sincerely,

Prof. Danielly C. Ferraz da Costa, Ph.D.

Round 2

Reviewer 3 Report

Comments and Suggestions for Authors

Despite the authors' effort the manuscript is not satisfactory

Comments on the Quality of English Language

none